# What are the prognostic factors for the development of incontinence-associated dermatitis (IAD): a protocol for a systematic review and meta-analysis

Julie Deprez [1,2] Jan Kottner [2,3] Alexandra Eilegård Wallin [1]
Nils Ohde [3] Carina Bååth [4,5] Ami Hommel [6] Lisa Hultin [7,8]
Anna Josefson [9,10] Dimitri Beeckman [1,2]

For numbered affiliations see end of article.

**Correspondence to**
Julie Deprez;
julie.deprez@oru.se

## ABSTRACT

**Introduction** Incontinence-associated dermatitis (IAD) is irritant contact dermatitis and skin damage associated with prolonged skin contact with urine and/or faeces. Identifying prognostic factors for the development of IAD may improve management, facilitate prevention and inform future research.

**Methods and analysis** This protocol follows the guidelines of the Preferred Reporting Items for Systematic Review and Meta-Analysis Protocols. Prospective and retrospective observational studies or clinical trials in which prognostic factors associated with the development of IAD are described are eligible. There are no restrictions on study setting, time, language, participant characteristics or geographical regions. Reviews, editorials, commentaries, methodological articles, letters to the editor, cross-sectional and case–control studies, and case reports are excluded. MEDLINE, CINAHL, EMBASE and The Cochrane Library will be searched from inception until May 2023. Two independent reviewers will independently evaluate studies. The Quality in Prognostic Studies tool will be used to assess the risk of bias, and the Checklist for Critical Appraisal and Data Extraction for Systematic Reviews of Prediction Modelling Studies-Prognostic Factors checklist will be used for data extraction of the included studies. Separate analyses will be conducted for each identified prognostic factor, with adjusted and unadjusted estimated measures analysed separately. Evidence will be summarised with a meta-analysis when possible, and narratively otherwise. The Q and $I^2$ statistics will be calculated in order to quantify heterogeneity. The quality of the evidence obtained will be evaluated according to the Grades of Recommendation Assessment, Development and Evaluation guidance.

**Ethics and dissemination** No ethical approval is needed since all data is already publicly accessible. The results of this work will be published in a peer-reviewed scientific journal.

## INTRODUCTION

Incontinence-associated dermatitis (IAD) is one form of irritant contact dermatitis (EK02.22) and describes cutaneous inflammation and skin damage associated with

prolonged skin contact with urine and/or faeces.[1 2]

### STRENGTHS AND LIMITATIONS OF THIS STUDY

⇒ This is the first systematic review to summarise the quality of evidence of prognostic factors for incontinence-associated dermatitis.
⇒ The risk of bias and the certainty of evidence will be evaluated using state-of-the-art methods for each prognostic factor separately.
⇒ Because of interaction and confounding the generalisability of the identified prognostic factors might be limited.

Clinical signs of IAD include skin erythema and oedema, occasionally with bullae and serous exudate, erosion or subsequent cutaneous infection.[3 4] Patients with IAD may experience discomfort, pain, burning, itching or tingling.[5] The presence of IAD can result in an excessive burden of care, loss of independence, disruption in activities and/or sleep, and a reduction in quality of life, which worsens with higher frequency and quantity of soiling.[1 6 7] Additionally, IAD is associated with an increased risk of secondary infections, and the development of pressure ulcers.[1 8–10] Managing all of these require additional resources, including prolonged hospital stays, resulting in a rise in healthcare expenses.[4 11]

Prevalence ranges between 5.2% and 46%, depending on the type of setting and population under study.[9 12]

By definition, incontinence is a prerequisite for having IAD, but in fact not all incontinent individuals develop IAD.[13] This suggests that many other factors and characteristics increase or decrease a person's susceptibility of developing IAD. Risk factors for IAD have been described,[14 15] but to our knowledge, a

Table 1   Key items for framing aim, search strategy, and study inclusion and exclusion criteria for systematic review, following PICOTS

|  | Definition |
|---|---|
| Population | Incontinent adults (>18 years) and adolescents (age 10–18 years)[32] Excluded: New borns, infants and young children (<10 years) |
| Index prognostic factor | Any variable that existed prior to the onset of IAD and assessed for association with IAD |
| Comparator prognostic factors | Any other variable that existed before the onset of IAD |
| Outcomes | 1.  Development of a new IAD 2.  Severity of IAD |
| Time | Minimum: 3 days Maximum: the duration of the study or observation period |
| Setting | Any healthcare setting (examples include but are not limited to: hospitals, nursing homes, community care…) |

IAD, incontinence associated dermatitis; PICOTS, Population, Index prognostic factor, Comparator prognostic factor, Outcome, Timing, Setting.

systematic review and evidence appraisal on prognostic factors has not yet been conducted. 'A prognostic factor is any variable that is associated with the risk of a subsequent health outcome among people with a particular health condition.' (Riley et al[16 p2]).

As such, our objective is to conduct a systematic review, and meta-analysis of prognostic factors associated with the development of IAD.

### Review question

What are prognostic factors associated with the development of IAD in incontinent patients?

### METHODS AND ANALYSIS

This protocol conforms to the Preferred Reporting Items for Systematic Review and Meta-Analysis Protocols checklist. The protocol will be registered with the PROSPERO international prospective register of systematic reviews. The review is scheduled to be conducted between May 2023 and May 2024.

### Eligibility criteria

Table 1 shows the description of the PICOTS (Population, Index prognostic factor, Comparator prognostic factor, Outcome, Timing, Setting) aspects.[16] Newborns, infants and children under 10 years of age will be excluded, due to differences in the structural and physiological properties of their skin compared with adults. For instance, the pH of neonatal and infant skin is typically higher, making them more susceptible to irritants and contributing to the development of conditions such as diaper

dermatitis. Additionally, the skin of newborns is thinner and more vulnerable than that of adults.[17] The following study designs are eligible: randomised controlled trials (RCTs), controlled clinical trials (CCTs), prospective and retrospective cohort studies. Reviews, editorials, commentaries, methodological articles, letters to the editor, cross-sectional studies, case–control studies and case reports, will be excluded. Case–control studies, although useful to identify prognostic factors, will be excluded due to the high risk of bias.[18] There are no restrictions on study setting, time, language, participant characteristics or geographical regions. Studies will be rejected if it is not clear from the publication that the predictors existed prior to the onset of IAD. Due to the variety of language used to describe incontinence-associated skin problems,[19 20] all variations of terms will be included. If they describe the skin's reactive response to repeated exposure to urine and/or faeces, which may appear as inflammation and erythema with or without erosion or denudation.[3 4 21]

### Information sources

To identify articles, the databases MEDLINE, CINAHL, EMBASE and The Cochrane Library will be searched. Additional relevant studies will be identified by cited and citing references. ClinicalTrials.gov (www.clinicaltrials.gov) and the WHO International Clinical Trials Registry Platform (apps.who.int/trialsearch/) will be searched for unpublished and ongoing studies. Articles will be electronically retrieved and in cases where electronic copies are unavailable, hard copies of these articles will be requested from the university library.

### Search strategy

A library technician with expertise in systematic reviews at the Örebro University, Örebro, Sweden has been consulted to develop the search methods. The search strategy consists of key terms combined with Boolean operators. Key terms are defined using the PICOTS and include incontinence associated dermatitis and prognostic factors using various combinations and synonyms. The Boolean operator OR is used to combine search terms for the same concept. The Boolean operator AND is used to combine the concepts. The search strategy will be adapted to each database. When appropriate MeSH terms or comparable alternatives will be used (see online supplemental appendix).

### Study records
#### Data management

Retrieved records will be imported in Endnote and duplicates will be removed using the duplicate search function and by manually reviewing the list. Articles will then be imported into Covidence,[22] an online tool that facilitates blind article screening by multiple reviewers.

#### Selection process

Titles and abstracts of all identified records will be reviewed for suitability by two researchers independently

**Table 2** Data charting variables and domains relating to article description

| Domain | Selected items and examples |
|---|---|
| General study information | Title, authors names, year of publication |
| Study characteristics | Sample size, study design, duration of follow-up, geographical location |
| Participant characteristics | Participant description, eligibility criteria |
| Setting | Type of care institution (hospital (unit), homecare, residential care…) |
| Definition of outcomes | IAD definition used by authors. Incontinence definition used by authors. |
| Prognostic factors | Examples include but are not limited to: presence of faecal incontinence and/or urinary incontinence, care dependency, level of personal hygiene… |
| Missing data | Attrition (quantification and reasons), handling of missing data by the study authors |
| Analysis | Univariate analysis/logistic regression/Cox regression/other |
| Results | Estimates reported between the prognostic factor and each review outcome: (A) unadjusted estimate, (B) adjusted estimates. Type of measure of association: OR, risk ratio (RR) or HR |

using the systematic review software Covidence.[22] For each excluded article, reasons for exclusion will be specified. Disagreement or doubt will be resolved by consensus and if consensus cannot be reached, a full-text screening will be conducted, and a third reviewer will be consulted. The full texts of the remaining articles will be individually assessed for eligibility by the same reviewers. Any doubts will be resolved by a third reviewer.

### Data collection process
The data extraction will be independently completed by two reviewers. Data extraction forms will be used to extract the relevant information and evidence. Any disagreement will be resolved through consultation with another reviewer.

### Data items
The Checklist for Critical Appraisal and Data Extraction for Systematic Reviews of Prediction Modelling Studies-Prognostic Factors[16] will guide data extraction. It has nine domains that cover information required to extract data in a reliable manner: source of data, participants, outcomes to be predicted, prognostic factors, sample size, missing data, analysis, results and interpretation and discussion. Table 2 shows a list of data items that will be extracted.

### Outcomes and prioritisation
The main predicted outcome will be the development of new IAD, defined as an irritant contact dermatitis resulting from prolonged contact of the skin with urine and/or faeces.[2] The severity of IAD will also be examined.

### Assessment of a risk of bias in individual studies
The Quality in Prognostic Factor Studies[23] tool will be used to assess the risk of bias of each of the prognostic factors in individual studies. It has six domains (study participants, study attrition, prognostic factor measurement, outcome measurement, study confounding and statistical analysis and reporting) with a series of prompting items for each domain. Each domain is assigned a risk of bias rating of high, moderate or low, and the overall risk of bias of a study is determined by a cumulative rating of all domains. The risk of bias will be independently assessed by two reviewers. Disagreements will be discussed with a third author until consensus is reached.

### Data synthesis
A table will be created summarising the characteristics of identified prognostic factors. Then, the strength and directions of association will be summarised for each prognostic factor separately. ORs, RRs and HRs will be used as measures of association. Mean differences will be considered for continuous prognostic factors. Adjusted and unadjusted estimates for each prognostic factor will be analysed separately. When at least five studies report similar measures of association, a meta-analysis will be conducted for each prognostic factor, if the measurement of this factor is comparable.[16 24] When neither OR, RR or HR nor corresponding SEs are reported, any available data from the included study, such as CIs, Kaplan-Meier curves, logrank test p values, will be used to estimate the associations if possible.[25 26] A random-effects model will be used when conducting meta-analysis, since it is assumed that there is a high degree of variability among studies due to clinical and methodological differences.[27] The effects of each prognostic factor will be estimated as an OR or standardised mean difference with a 95% CI and $T^2$, which indicates between-study variance. Furthermore, a 95% prediction interval will also be calculated.[28] The heterogeneity between studies in each meta-analysis will be assessed visually by examining forest plots and calculating the Q and $I^2$ statistics.[29] When it is not feasible to conduct a quantitative synthesis, the evidence will be summarised narratively.

### Bias
The presence of small study bias, including publication bias, will be evaluated graphically by examining the presence of asymmetry in a funnel plot. Additionally, the Egger's test will be used to make a statistical assessment if a meta-analysis is based on 10 or more studies for an outcome.[30] For this test, the natural logarithmic scale of the OR must be regressed against its SE, and statistical significance for asymmetry is set at the 10% level due to

the low power of the test. If publication bias is statistically suspected, the method of trim and fill can be used to estimate the number of missing studies and provide an adjusted summary effect.[31] To assess selective reporting, the consistency of study findings with its protocol must be examined, if available.

## Confidence in cumulative evidence

Using the modified Grading of Recommendations, Assessment, Development and Evaluation (GRADE) approach, the quality of evidence for each prognostic factor will be evaluated. The certainty rating for observational studies will begin at high, while the secondary analysis of highly controlled RCTs will start at low. The domains of risk of bias, inconsistency, indirectness, imprecision, publication bias, large effect, dose response and plausible confounding will be taken into account when deciding to downgrade or upgrade the evidence's certainty.[18]

## Patient and public involvement

No patients or service users were involved in the design of this review protocol.

## ETHICS AND DISSEMINATION

This systematic review does not require ethics approval as it will be a summary of previously published evidence. No individual patient data will be obtained or accessed. Therefore, there is no concerning ethical issue in the conduct of this research. Following the completion of the systematic review, findings will be presented to academic audiences at international conferences focusing on skin integrity. The results will also be published in a peer-reviewed academic journal to reach clinical and academic experts interested in the topic. Plain language summaries and presentations to hospitals and other relevant clinical programmes will also be developed.

**Author affiliations**
[1]Swedish Centre for Skin and Wound Research (SCENTR), School of Health Sciences, Faculty of Medicine and Health, Örebro University, Örebro, Sweden
[2]Skin Integrity Research Group (SKINT), University Centre for Nursing and Midwifery, Department of Public Health and Primary Care, Faculty of Medicine and Health Sciences, Ghent University, Ghent, Belgium
[3]Institute of Clinical Nursing Science, Charité Universitätsmedizin, Berlin, Germany
[4]Department of Health Sciences, Faculty of Health, Science and Technology, Karlstad University, Karlstad, Sweden
[5]Faculty of Health, Welfare and Organisation, Østfold University College - Campus Frederikstad, Fredrikstad, Norway
[6]Department of Care Science, Malmö University, Malmö, Sweden
[7]Department of Public Health and Caring Sciences, Upsalla University, Upsalla, Sweden
[8]Upsalla University Hospital, Upsalla, Sweden
[9]School of Medical Sciences, Faculty of Medicine and Health, Örebro University, Örebro, Sweden
[10]Department of Dermatology, Örebro University Hospital, Örebro, Sweden

**Acknowledgements** We acknowledge the support and funding from Vetenskapsrådet.

**Contributors** JK, AEW, DB, CB, LH, AJ and AH provided the idea of the topic. All authors contributed to the conception of the research question. JD designed and wrote the protocol in close collaboration with JK, AEW, NO and DB. All authors contributed to the development of search strategies, eligibility criteria and methodology for data synthesis. All authors revised the protocol critically for important intellectual content. All authors contributed and provided feedback to the draft protocol and approved the final version of this protocol. JD and NO will review the titles and abstracts of all studies obtained using the search strategy to exclude articles that do not meet the eligibility criteria.

**Funding** This work was supported by Vetenskapsrådet, grant number 2021-02653.

**Competing interests** None declared.

**Patient and public involvement** Patients and/or the public were not involved in the design, or conduct, or reporting, or dissemination plans of this research.

**Patient consent for publication** Not applicable.

**Provenance and peer review** Not commissioned; externally peer reviewed.

**ORCID iDs**
Julie Deprez http://orcid.org/0000-0002-8396-6761
Jan Kottner http://orcid.org/0000-0003-0750-3818
Alexandra Eilegård Wallin http://orcid.org/0000-0002-6133-8975
Nils Ohde http://orcid.org/0009-0001-5200-1019
Carina Bååth http://orcid.org/0000-0002-9608-336X
Ami Hommel http://orcid.org/0000-0002-6114-6535
Lisa Hultin http://orcid.org/0000-0001-8270-8560
Anna Josefson http://orcid.org/0000-0002-7478-056X
Dimitri Beeckman http://orcid.org/0000-0003-3080-8716

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
