## [Reviewer comments · BMJ Open]

ARTICLE DETAILS

TITLE (PROVISIONAL)	What are the prognostic factors for the development of incontinence-associated dermatitis (IAD): A protocol for a Systematic Review and Meta-Analysis
AUTHORS	Deprez, Julie; Kottner, Jan; Eilegård Wallin, Alexandra; Ohde, Nils; Bååth, Carina; Hommel, Ami; Hultin, Lisa; Josefson, Anna; Beeckman, Dimitri

VERSION 1 – REVIEW

REVIEWER	Kayser, Susan Baxter Healthcare Corp, HEOR I am employed by Baxter, which makes an incontinence detection device
REVIEW RETURNED	14-Mar-2023

GENERAL COMMENTS	Hello, Thank you for the well organized and well written protocol. Everything seems in order and well thought through and I appreciate you submitting your protocol for review. I have only one minor comment. In your study inclusion criteria you specify you will examine variables "collected prior to the diagnosis of IAD." In my experience reading manuscripts on factors that are associated with IAD, some do not specify whether the factor was or was not collected prior to the diagnosis because timestamps are not always available in the databases. How will you handle papers that have ambiguous methods or use retrospective databases where the specific time of IAD diagnosis may not be recorded? Thank you and I look forward to having the opportunity to read the manuscript when it's published.
---

REVIEWER	Mugita, Yuko University of Tokyo Graduate School of Medicine School of Medicine Department of Physiology
REVIEW RETURNED	24-Mar-2023

GENERAL COMMENTS	I believe this is a review of a very important topic that can identify the prognostic factors of IAD. Please refer to the comments below and consider making additional revisions. 1. Abstract Reading the abstract alone, it appears to include prospective observational studies but not backward-looking observational studies. Since the text states that it includes backward-looking observational studies, please do the same in the abstract.
---

	2. Methods This review includes observational studies but excludes case-control studies. Please state the reason why they were excluded. Case-control studies would also be appropriate for the review question if the purpose was to explore the causes of IAD occurrence. 3. Methods Please describe the reasons for excluding neonates and children under 10 years of age from the target population. From the review question, it does not seem necessary to limit the target population.
--	---

VERSION 1 – AUTHOR RESPONSE

Reviewer #1: Dr. Susan Kayser, Baxter Healthcare Corp

Hello,

Thank you for the well organized and well written protocol. Everything seems in order and well thought through and I appreciate you submitting your protocol for review. I have only one minor comment. In your study inclusion criteria you specify you will examine variables "collected prior to the diagnosis of IAD." In my experience reading manuscripts on factors that are associated with IAD, some do not specify whether the factor was or was not collected prior to the diagnosis because timestamps are not always available in the databases. How will you handle papers that have ambiguous methods or use retrospective databases where the specific time of IAD diagnosis may not be recorded?

Thank you and I look forward to having the opportunity to read the manuscript when it's published.

• Response:

Dear dr. Susan Kayser,

Thank you for taking the time to review our article. We appreciate your valuable feedback and relevant comments. You are correct that the timing of when the potential factors/variables were collected may not always be clear or available in some studies. However, our intention with the inclusion criteria was to ensure that the identified factors were present prior to the development of IAD, regardless of when they were collected. If it is not clear from the publication that the predictors existed before the onset of IAD, the work will be excluded. To make this clearer, we added it to the methods section (see p. 4, line 133-134, marked in yellow) and updated the inclusion and exclusion criteria table (see p. 4, Table 1, marked in yellow) Thank you for bringing this to our attention.

Reviewer #2: Dr. Yuko Mugita, University of Tokyo Graduate School of Medicine School of Medicine Department of Physiology

I believe this is a review of a very important topic that can identify the prognostic factors of IAD. Please refer to the comments below and consider making additional revisions.

Dear dr. Yuko Mugita,

Thank you for taking the time to review our article. We appreciate your valuable feedback and comments.

1. Abstract

Reading the abstract alone, it appears to include prospective observational studies but not backward-looking observational studies. Since the text states that it includes backward-looking observational studies, please do the same in the abstract.

• Response: We have updated the abstract to specify that retrospective cohort studies will also be included, which was not previously clear. We thank you for this comment (see p.2, line 54, marked in yellow).

2. Methods

This review includes observational studies but excludes case-control studies. Please state the reason why they were excluded. Case-control studies would also be appropriate for the review question if the purpose was to explore the causes of IAD occurrence.

- Response: We agree that case-control studies are useful for investigating the causes of IAD. Our primary goal is to identify prognostic factors associated with the development of IAD in incontinent patients in real-world settings. Given the high risk of bias in case-control studies, we decided to exclude these studies. This was added to the method section (see p. 4, line 131-132, marked in yellow).

3. Methods

Please describe the reasons for excluding neonates and children under 10 years of age from the target population. From the review question, it does not seem necessary to limit the target population.

- Response: We decided to exclude neonates and children under 10 years of age from the target population due to differences in the structural and physiological properties of their skin compared to adults.

We added this to the method section of the manuscript (see p. 3-4, line 124-128, marked in yellow).